# FROM PLAY TO POLICY: CONDITIONAL BEHAVIOR GENERATION FROM UNCURATED ROBOT DATA

**Zichen Jeff Cui**[*]      **Yibin Wang**      **Nur Muhammad (Mahi) Shafiullah**      **Lerrel Pinto**

New York University

## ABSTRACT

While large-scale sequence modeling from offline data has led to impressive performance gains in natural language and image generation, directly translating such ideas to robotics has been challenging. One critical reason for this is that uncurated robot demonstration data, i.e. *play* data, collected from non-expert human demonstrators are often noisy, diverse, and distributionally multi-modal. This makes extracting useful, task-centric behaviors from such data a difficult generative modeling problem. In this work, we present Conditional Behavior Transformers (C-BeT), a method that combines the multi-modal generation ability of Behavior Transformer with future-conditioned goal specification. On a suite of simulated benchmark tasks, we find that C-BeT improves upon prior state-of-the-art work in learning from play data by an average of $45.7\%$. Further, we demonstrate for the first time that useful task-centric behaviors can be learned on a real-world robot purely from play data without any task labels or reward information. Robot videos are best viewed on our project website: `play-to-policy.github.io`.

## 1 INTRODUCTION

Machine Learning is undergoing a Cambrian explosion in large generative models for applications across vision (Ramesh et al., 2022) and language (Brown et al., 2020). A shared property across these models is that they are trained on large and uncurated data, often scraped from the internet. Interestingly, although these models are trained without explicit task-specific labels in a self-supervised manner, they demonstrate a preternatural ability to generalize by simply conditioning the model on desirable outputs (e.g. "prompts" in text or image generation). Yet, the success of conditional generation from uncurated data has remained elusive for decision making problems, particularly in robotic behavior generation.

To address this gap in behavior generation, several works (Lynch et al., 2019; Pertsch et al., 2020b) have studied the use of generative models on *play* data. Here, play data is a form of offline, uncurated data that comes from either humans or a set of expert policies interacting with the environment. However, once trained, many of these generative models require significant amounts of additional online training with task-specific rewards (Gupta et al., 2019; Singh et al., 2020). In order to obtain task-specific policies without online training, a new line of approaches employ offline RL to learn goal-conditioned policies (Levine et al., 2020; Ma et al., 2022). These methods often require rewards or reward functions to accompany the data, either specified during data collection or inferred through hand-crafted distance metrics, for compatibility with RL training. Unfortunately, for many real-world applications, data does not readily come with rewards. This prompts the question: *how do we learn conditional models for behavior generation from reward-free, play data?*

Table 1: Comparison between existing algorithms to learn from large, uncurated datasets: GCBC (Lynch et al., 2019), GCSL (Ghosh et al., 2019), Offline GCRL (Ma et al., 2022), Decision Transformer Chen et al. (2021)

|  | GCBC | GCSL | Offline RL | Decision Transformer | C-BeT (ours) |
|---|---|---|---|---|---|
| Reward-free | ✓ | ✓ | ✗ | ✗ | ✓ |
| Offline | ✓ | ✗ | ✓ | ✓ | ✓ |
| Multi-modal | ✗ | ✗ | ✗ | ✗ | ✓ |

[*]Corresponding author, email: `jeff.cui@nyu.edu`

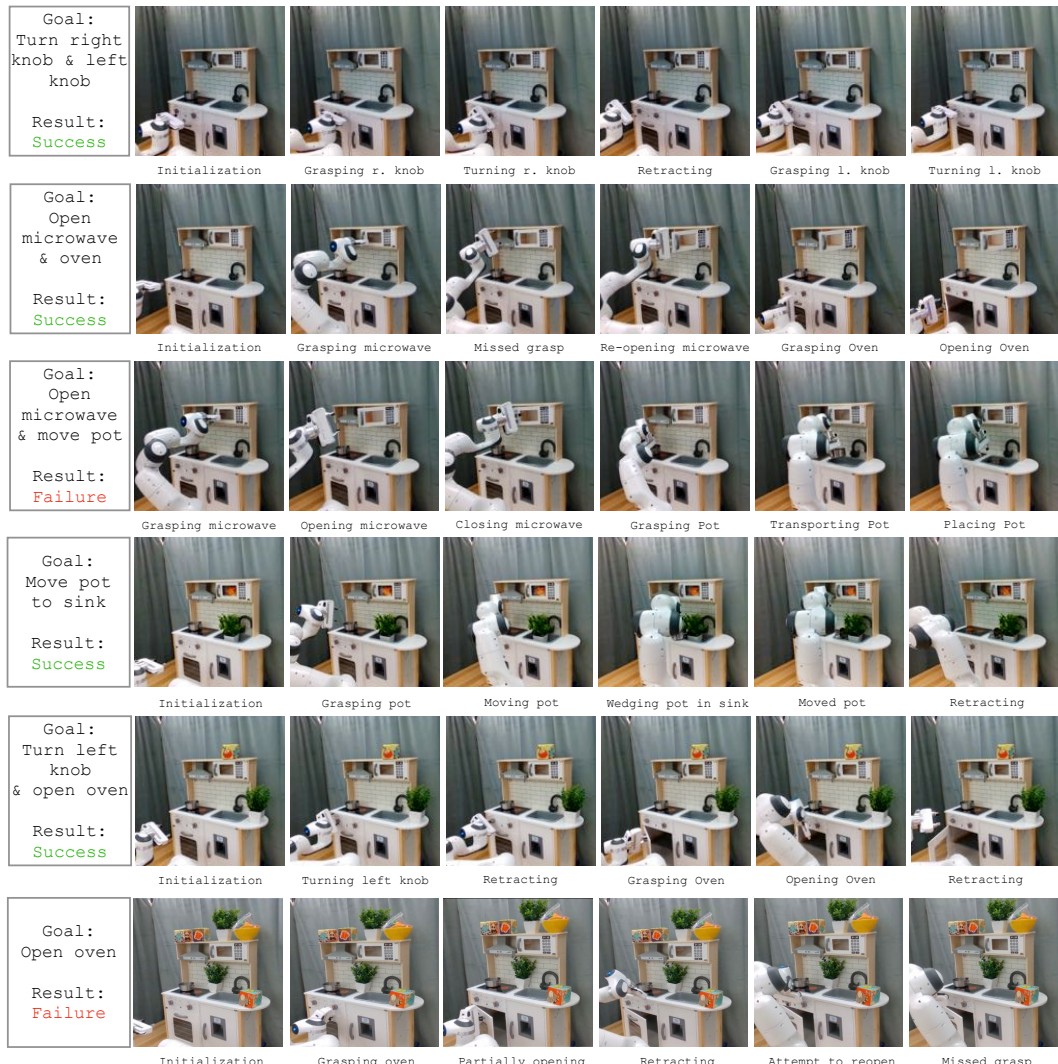

Figure 1: Multiple conditioned roll-outs of visual robot policies learned on our toy kitchen with only 4.5 hours of human play interactions. Our model learns purely from image and proprioception without human labeling or data curation. During evaluation, the policy can be conditioned either on a goal observation or a demonstration. Note that the last three rows contain distractor objects in the environment that were never seen during training.

To answer this question, we turn towards transformer-based generative models that are commonplace in text generation. Here, given a prompt, models like GPT-3 (Brown et al., 2020) can generate text that coherently follow or satisfy the prompt. However, directly applying such models to behavior generation requires overcoming two significant challenges. First, unlike the discrete tokens used in text generation, behavior generation will need models that can output continuous actions while also modeling any multi-modality present in the underlying data. Second, unlike textual prompts that serve as conditioning for text generation, behavior generation may not have the condition and the operand be part of the same token set, and may instead require conditioning on future outcomes.

In this work, we present Conditional Behavior Transformers (C-BeT), a new model for learning conditional behaviors from offline data. To produce a distribution over continuous actions instead of discrete tokens, C-BeT augments standard text generation transformers with the action discretization introduced in Behavior Transformers (BeT) (Shafiullah et al., 2022). Conditioning in C-BeT is done by specifying desired future states as input similar to Play-Goal Conditioned Behavior Cloning (Play-GCBC) (Lynch et al., 2019). By combining these two ideas, C-BeT is able to leverage the multi-modal generation capabilities of transformer models with the future conditioning capabilities of conditional policy learning. Importantly, C-BeT does not require any online environment interactions during training, nor the specification of rewards or Q functions needed in offline RL.

We experimentally evaluate C-BeT on three simulated benchmarks (visual self-driving in CARLA (Dosovitskiy et al., 2017), multi-modal block pushing (Florence et al., 2021), and simulated kitchen (Gupta et al., 2019)), and on a real Franka robot trained with play data collected by human volunteers. The main findings from these experiments can be summarized as:

1. On future-conditioned tasks, C-BeT achieves significantly higher performance compared to prior work in learning from play.

2. C-BeT demonstrates that competent visual policies for real-world tasks can be learned from fully offline multi-modal play data (rollouts visualized in Figure 1).

## 2 BACKGROUND AND PRELIMINARIES

**Play-like data:** Learning from Demonstrations (Argall et al., 2009) is one of the earliest frameworks explored for behavior learning algorithms from offline data. Typically, the datasets used in these frameworks have a built in assumption that the demonstrations are collected from an expert repeatedly demonstrating a single task in exactly the same way. On the contrary, play datasets violate many of such assumptions, like those of expertise of the demonstrator, and the unimodality of the task and the demonstrations. Algorithms that learn from such datasets sometimes assume that the demonstrations collected are from a rational agent with possibly some latent intent in their behavior (Lynch et al., 2019). Note that, unlike standard offline-RL datasets (Fu et al., 2020), play-like behavior datasets neither contain fully random behaviors, nor have rewards associated with the demonstrations.

**Behavior Transformers (BeT):** BeT (Shafiullah et al., 2022) is a multi-modal behavior cloning model designed particularly for tackling play-like behavior datasets. BeT uses a GPT-like transformer architecture to model the probability distribution of action given a sequence of states $\pi(a_t \mid s_{t-h:t})$ from a given dataset. However, unlike previous behavior learning algorithms, BeT does not assume a unimodal prior for the action distribution. Instead, it uses a $k$-means discretization to bin the actions from the demonstration set into $k$ bins, and then uses the bins to decompose each action into a discrete and continuous component. This support for multi-modal action distributions make BeT particularly suited for multi-modal, play-like behavior datasets where unimodal behavior cloning algorithms fail. However, vanilla BeT only supports unconditonal behavior rollouts, which means that it is not possible to choose a targeted mode of behavior during BeT policy execution.

**Conditional behavior learning:** Generally, the problem of behavior learning for an agent is considered the task of learning a *policy* $\pi : \mathcal{O} \to \mathcal{A}$ mapping from the environment observations to the agent's actions that elicit some desired behavior. Conditional behavior learning is concerned with learning a policy $\pi : \mathcal{O} \times \mathcal{G} \to \mathcal{A}$ conditioned additionally on a secondary variable $g$ sampled from a distribution $p(g)$. This condition variable could be specific environment states, latents (such as one-hot vectors), or even image observations. The success of a conditioned policy can be evaluated either through pre-specified reward functions, distance function between achieved outcome $g'$ and specified outcome $g$, or by discounted visitation probability $d_{\pi(\cdot|g)} = \mathbb{E}_{\tau \sim \pi}[\sum_{t=0}^{\infty} \gamma^t \delta(\phi(o_t) = g)]$ if a mapping $\phi$ between states and achieved outcome is defined (Eysenbach et al., 2022).

**Goal Conditioned Behavior Cloning (GCBC):** In GCBC (Lynch et al., 2019; Emmons et al., 2021), the agent is presented with a dataset of (observation, action, goal) tuples $(o, a, g)$, or sequences of such tuples, and the objective of the agent is to learn a goal-conditioned behavior policy. The simplest way to achieve so is by training a policy $\pi(\cdot \mid o, g)$ that maximizes the probability of the seen data $\pi^* = \arg\max_\pi \prod_{(o,a,g)} \mathbb{P}[a \sim \pi(\cdot \mid o, g)]$. Assuming a unimodal Gaussian distribution for $\pi(a \mid o, g)$ and a model parametrized by $\theta$, this comes down to finding the parameter $\theta$ minimizing the MSE loss, $\theta^* = \arg\min_\theta \sum_{(o,a,g)} ||a - \pi(o, g; \theta)||^2$. To make GCBC compatible with play data that inherently does not have goal labels, goal relabeling from future states is often necessary. A common form of data augmentation in training such models, useful when $\mathcal{G} \subset \mathcal{O}$, is hindsight data relabeling (Andrychowicz et al., 2017), where the dataset $\{(o, a, g)\}$ is augmented with $\{(o_t, a, o_{t'}) \mid t' > t\}$ by relabeling any reached state in a future timestep as a goal state and adding it to the dataset.

## 3    APPROACH

Given a dataset $\{(o, a)\} \in \mathcal{O} \times \mathcal{A}$ of sequences of (observation, action) pairs from a play dataset, our goal is to learn a behavior generation model that is capable of handling multiple tasks and multiple ways of accomplishing each task. At the same time, we wish to be able to extract desired behavior from the dataset in the form of a policy through our model, or, in terms of generative models, "controllably generate" our desired behavior (see Figure 2). Finally, in the process of learning this controllable, conditional generative model, we wish to minimize the amount of additional human annotation or curation required in preparing the dataset. The method we develop to address these needs is called Conditional Behavior Transformer.

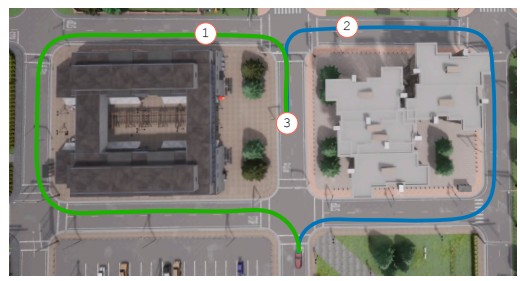

Figure 2: Conditional behavior learning from play demonstrations. Here, a policy conditioned on reaching ① or ② has only one possible course of action, but conditioned on reaching ③ there are two reasonable paths.

### 3.1    CONDITIONAL BEHAVIOR TRANSFORMERS (C-BeT)

**Conditional task formulation:** First, we formulate the task of learning from a play dataset as learning a conditional behavior policy, i.e. given the current state, we need to model the distribution of actions that can lead to particular future states. For simplicity, our formulation can be expressed as $\pi : \mathcal{O} \times \mathcal{O} \to \mathcal{D}(\mathcal{A})$ where, given a current observation $o_c$ and a future observation $o_g$, our policy $\pi$ models the distribution of the possible actions that can take the agent from $o_c$ to $o_g$. Mathematically, given a set of play trajectories $T$, we model the distribution $\pi(a \mid o_c, o_g) \triangleq \mathbb{P}_{\tau \in T}(a \mid o_c = \tau_t, o_g = \tau_{t'}, t' > t)$. Next, to make our policy more robust since we operate in the partially observable setting, we replace singular observations with a sequence of observations; namely replacing $o_c$ and $o_g$ with $\bar{o}_c = o_c^{(1:N)}$ and $\bar{o}_g = o_g^{(1:N)}$ for some integer $N$. Thus, the final task formulation becomes learning a generative model $\pi$ with:

$$\pi\left(a \mid o_c^{(1:N)}, o_g^{(1:N)}\right) \triangleq \mathbb{P}_{\tau \in T}\left(a \mid o_c^{(1:N)} = \tau_{t:t+N}, o_g^{(1:N)} = \tau_{t':t'+N}, t' > t\right) \quad (1)$$

**Architecture selection:** Note that the model for our task described in the previous paragraph is necessarily multi-modal, since depending on the sequences $\bar{o}_c$ and $\bar{o}_g$, there could be multiple plausible sequences of actions with non-zero probability mass. As a result, we choose Behavior Transformers (BeT) (Shafiullah et al., 2022) as our generative architecture base as it can learn action generation with multiple modes. We modify the input to the BeT to be a concatenation of our future conditional observation sequence and current observation sequence. We choose to concatenate the inputs instead of stacking them, as this allows us to independently choose sequence lengths for the current and future conditional observations. Since BeT is a sequence-to-sequence model, we only consider the actions associated with the current observations as our actions. We show the detailed architecture of our model in Figure 3.

**Dataset preparation:** To train a C-BeT model on our play dataset $\{(o, a)\}$, we will need to appropriately prepare the dataset. We first convert the dataset to hold sequences of observations associated with actions, $\{(o_{t:t+N}, a_{t:t+N})\}$. Then, during training time, we dynamically augment each pair with a sequence of future observations, functionally converting our dataset into $\{(o_{t:t+N}, a_{t:t+N}, o_{t':t'+N'})\}$ for some $t' > t$, and treat the sequence $o_{t':t'+N'}$ as $\bar{o}_g$.

**Training objective:** We employ the same objective as BeT in training C-BeT. For each of the current observation and future conditional pair, we compute the BeT loss (see appendix B for details) between the ground truth actions and the predicted actions. We compute the focal loss (Lin et al., 2017) on the predicted action bins, and the MT-loss (Girshick, 2015) on the predicted action offsets corresponding to the action bins as described in BeT.

**Test-time conditioning with C-BeT:** During test time, we again concatenate our future conditional sequence with our current observations, and sample actions from our model according to the BeT framework. While in this work, we primarily condition C-BeT on future observations, we also study

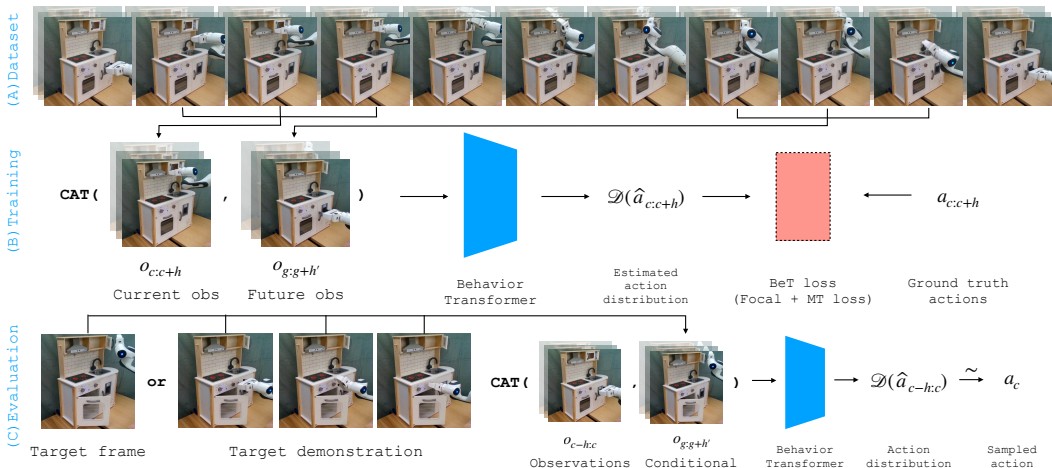

Figure 3: End-to-end training and evaluation of C-BeT. (A) Our dataset consists of play data in an environment, which may contain semi-optimal behavior, multi-modal demonstrations, and failures, and does not contain any annotations or task labels. (B) We train our C-BeT model by conditioning on current and future states using BeT (Section 2) (C) During evaluation, our algorithm can be conditioned by target observations or newly collected demonstrations to generate targeted behavior.

other ways of training and conditioning it, such as binary latent vectors denoting the modes in a trajectory in our experiments, and compare its performance to observation-conditioned C-BeT (see Section 4.5).

# 4 C-BeT ON SIMULATED BENCHMARKS

In this section, we discuss our experiments in simulation that are designed to answer the following key questions: How well does C-BeT learn behaviors from play? How important is multi-modal action modeling? And finally, how does C-BeT compare to other forms of conditioning?

## 4.1 BASELINES

We compare with the following state-of-the-art methods in learning from reward-free offline data:

- **Goal Conditioned BC (GCBC):** GCBC (Lynch et al., 2019; Emmons et al., 2021) learns a policy by optimizing the probability of seen actions given current and the end state in a trajectory.
- **Weighted Goal Conditioned Supervised Learning (WGCSL) (Yang et al., 2022):** GCSL (Ghosh et al., 2019) is an online algorithm with multiple rounds of collecting online data, relabeling, and training a policy on that data using GCBC. WGCSL (Yang et al., 2022) improves GCSL by learning an additional value function used to weight the GCSL loss. We compare against an single-round, offline variant of WGCSL in this work.
- **Learning Motor Primitives from Play (Play-LMP):** Play-LMP (Lynch et al., 2019) is a behavior generation algorithm that focuses on learning short ($\sim 30$ timesteps) motor primitives from play data. Play-LMP does so by using a variational-autoencoder (VAE) to encode action sequences into motor program latents and decoding actions from them.
- **Relay Imitation Learning (RIL):** Relay Imitation Learning (Gupta et al., 2019) is a hierarchical imitation learning with a high level controller that generates short term target state given long term goals, and a low level controller that generates action given short term target.
- **Conditional Implicit Behavioral Cloning (C-IBC):** Implicit behavioral cloning (Florence et al., 2021) learns an energy based model (EBM) $E(a \mid o)$ over demos and during test samples action $a$ given an observation $o$. We compare against a conditional IBC by training an EBM $E(a \mid o, g)$.
- **Generalization Through Imitation (GTI):** GTI (Mandlekar et al., 2020) encodes the goal condition using a CVAE, and autoregressively rolls out action sequences given observation and goal-latent. We follow their architecture and forgo collecting new trajectories with an intermediate model since that does not fit an offline framework.
- **Offline Goal-Conditioned RL:** While offline RL is generally incompatible with play data without rewards, recently some offline goal-conditioned RL algorithms achieved success by optimizing for

Table 2: Results of future-conditioned algorithms on a set of simulated environments. The numbers reported for CARLA, BlockPush, and Kitchen are out of 1, 1, and 4 respectively, following Shafiullah et al. (2022). In CARLA, success counts as reaching the location corresponding to the observation; for BlockPush, it is pushing one or both blocks into the target squares; and for Kitchen, success corresponds to the number of conditioned tasks, out of four, completed successfully.

| | GCBC | WGCSL | Play-LMP | RIL | C-IBC | GTI | GoFAR | BeT | C-BeT (unimodal) | C-BeT (multimodal) |
|---|---|---|---|---|---|---|---|---|---|---|
| CARLA | 0.04 | 0.02 | 0.0 | 0.59 | 0.65 | 0.74 | 0.72 | 0.31 | 0.62 | **0.98** |
| BlockPush | 0.06 | 0.10 | 0.02 | 0.07 | 0.01 | 0.04 | 0.04 | 0.34 | 0.35 | **0.90** |
| Kitchen | 0.74 | 1.17 | 0.04 | 0.39 | 0.13 | 1.61 | 1.24 | 1.77 | 2.74 | **2.80** |

a proxy reward defined through state occupancy. Our baseline, GoFAR (Ma et al., 2022), is one such algorithm that learns a goal-conditioned value function and optimizes a policy to maximize it.

- **Behavior Transformers (BeT):** We include unconditional BeT (Sec. 2) in our baseline to understand the improvements made by the C-BeT conditioning. In practice, it acts as a "random" baseline that performs the tasks without regard for the goal.

- **Unimodal C-BeT:** We use our method without the multi-modal head introduced in BeT. This also corresponds to a variant of Decision Transformer conditioning on outcomes instead of rewards.

Note that neither WGCSL nor GoFAR are directly compatible with image states and goals, since they require a proxy reward function $r : \mathcal{S} \times \mathcal{G} \to \mathbb{R}$. Thus, we had to design a proxy reward function on the image representations, $\exp\left(-(1/4||g - s||)^2\right)$ to apply them on image-based environments. For a fair comparison, we also upgrade baseline Play-LMP, C-IBC, and GTI architectures by giving them sequences of observations and retrofitting them with transformers whenever applicable.

## 4.2 SIMULATED ENVIRONMENTS AND DATASETS

We run our algorithms and baselines on a collection of simulated environments as a benchmark to select the best algorithms to run on our real robotic setup. The simulated environments are selected to cover a variety of properties that are necessary for the real world environment, such as pixel-based observations, diverse modes in the play dataset, and complex action spaces (see Figure. 4).

1. **CARLA self-driving:** CARLA (Dosovitskiy et al., 2017) is a simulated self-driving environment created using Unreal Engine. In this environment, the observations are RGB pixel values of dimension $(224, 224, 3)$, and actions are two-dimensional (accelerate/brake and steer). We use an environment with a fork in the road (see Figure 2) following two possible routes to the same goal, collecting 200 demonstrations in total. We condition on one of the two possible routes to the goal, and at the goal where choosing either of the two modes is valid.

2. **Multi-modal block-pushing:** We use the multi-modal block-pushing environment from Florence et al. (2021) for complicated multi-modal demonstrations. In this environment, an xArm robot pushes two blocks, red and green, into two square targets colored red and green. All positions are randomized with some noise at episode start. We use 1,000 demonstrations collected using a deterministic controller, and condition on just the future block positions on each baseline.

3. **Franka relay kitchen:** Originally introduced in Gupta et al. (2019), Relay Kitchen is a robotic environment in a simulated kitchen with seven possible tasks. A Franka Panda robot is used to manipulate the kitchen, and the associated dataset comes with 566 demonstrations collected by humans with VR controllers performing four of the seven tasks in some sequence.

## 4.3 HOW WELL DOES C-BET LEARN BEHAVIORS FROM PLAY?

On each of these environments, we train conditional behavior generation models and evaluate them on a set of conditions sampled from the dataset. The success is defined by the model performing the same tasks as conditioned by the future outcome. We see from Table. 2 that C-BeT performs significantly better compared to the baselines on all three tasks. BeT, as our unconditioned "random" baseline, shows the success rate of completing tasks unconditionally, and see that none of the baselines surpasses it consistently. Out of the MLP-based baselines, WGCSL performs best in the state-based tasks. However, GoFAR performs best on the CARLA vision based environment where the other two MLP-based baselines fail almost completely. We note that Play-LMP performs poorly because our tasks are long-horizon and quite far from its intended motor primitive regime, which may be challenging for Play-LMP's short-horizon auto-encoding architecture.

### 4.4 How important is multi-modal action modeling?

While we use a multi-modal behavior model in this work, it is not immediately obvious that it may be necessary. Specifically, some previous outcome-conditioned policy learning works (Chen et al., 2021; Emmons et al., 2021) implicitly assume that policies are unimodal once conditioned on an outcome. In Table 2 the comparison between C-BeT and unimodal C-BeT shows that this assumption may not be true for all environments, and all else being equal, having an explicitly multi-modal model helps learning an outcome conditioned policy when there may be multiple ways to achieve an outcome.

### 4.5 How does C-BeT compare to other forms of conditioning?

We consider the question of how much comparative advantage there is in getting human labels for our tasks. We do so by adding manual one-hot (CARLA, BlockPush) or binary (Kitchen) labels to our tasks, and training and evaluating C-BeT with those labels. As we see on Table 3, on the three simulated environments, C-BeT conditioned on only future observations performs comparably to conditioning with human labels.

Table 3: Comparison between C-BeT with no supervised labels and labels acquired with human supervision.

|  | No labels | Labels |
|---|---|---|
| CARLA | 0.98 | 1.0 |
| BlockPush | 0.90 | 0.89 |
| Kitchen | 2.80 | 2.75 |

## 5 C-BeT on Real-World Robotic Manipulation

We now discuss our robot experiments, which are geared towards understanding the usefulness of C-BeT on real-world play data.

### 5.1 Robotic Environment and Dataset

**Robot setup:** Our environment consists of a Franka Emika Panda robot, similar to the simulated Franka Kitchen environment, set up with a children's toy kitchen set (see Figure 1). The toy kitchen has an oven, a microwave, a pot, and two stove knobs that are relevant to our play dataset. The action space in this environment contains the seven joint angle deltas normalized within the $[-1, 1]$ range, and a binary gripper control.

**Play dataset:** We collected 460 sequences totaling to 265 minutes (about 4.5 hours) of play data on the toy kitchen with volunteers using a Vive VR controller to move the Franka. While collecting the play data, we did not give the volunteers any explicit instructions about doing any particular tasks, or number of tasks, beyond specifying the interactable items, and stipulating that the pot only goes on the left stove or in the sink, to prevent dropping the pot and reaching an unresettable state. As the observations, we save the RGB observations from two cameras on the left and right of the setup, as well as the robot's proprioceptive joint angles. Overall, the dataset contains 45 287 frames of play interactions and their associated actions.

**Representation learning** To simplify the task of learning policies on image space, we decouple the task of image representation learning from policy learning following Pari et al. (2021). For each camera, we first fine-tune a pretrained ResNet-18 (He et al., 2016) encoder on the acquired frames with BYOL self-supervision (Grill et al., 2020). Then, during policy learning and evaluation, instead of the image from the cameras, we pass the two 512-dimensional BYOL embeddings as part of the observation. For the proprioceptive part of the observation, we repeat the $(\sin, \cos)$ of seven joint states 74 times to get a 1036-dimensional proprioceptive representation, making our overall observation representation 2060-dimensional.

### 5.2 Conditional Behavior Generation on Real Robot

**Behavior generation on single tasks:** Our first experiment in the real robot is about extracting single-task policies from the play dataset. We define our tasks as manipulating the four types of interactable objects one at a time: opening the oven door, opening the microwave door, moving the pot from the stove to the sink, and rotating a knob 90 degrees to the right. We use appropriate conditioning frames from our observation dataset, and start the robot from the neutral state to complete the four tasks. The result of this experiment is presented in Table 4. We see that on single task conditionals,

Table 4: Single-task success rate in a real world kitchen with conditional models. We present the success rate and number of trials on each task, with cumulative results presented on the last column.

|  | Knobs | Oven | Microwave | Pot | Cumulative |
|---|---|---|---|---|---|
| GoFAR | 0/10 | 0/5 | 0/5 | 0/5 | 0/25 |
| Unconditional BeT | 5/20 | 6/10 | 1/10 | 0/10 | 12/50 |
| Unimodal C-BeT | 1/20 | 8/10 | 4/10 | 0/10 | 13/50 |
| Multimodal C-BeT | 3/20 | 9/10 | 7/10 | 5/10 | 24/50 |

Table 5: Task success rate in a real world kitchen with conditional models evaluated on a long-horizon goal. We present the success rate and number of trials on each task, with cumulative result presented on the last column.

|  | Oven $\rightarrow$ Pot | Microwave $\rightarrow$ Oven | Pot $\rightarrow$ Microwave | Avg. Tasks/Run |
|---|---|---|---|---|
| Unconditional BeT | $(6, 0)/10$ | $(1, 6)/10$ | $(0, 1)/10$ | 0.47 |
| Unimodal C-BeT | $(1, 1)/10$ | $(2, 0)/10$ | $(8, 0)/10$ | 0.37 |
| Multimodal C-BeT | $(5, 4)/10$ | $(8, 8)/10$ | $(4, 4)/10$ | 1.1 |

C-BeT is able to complete all tasks except the knobs consistently, outperforming all our baselines, showing that C-BeT is able to extract single-task policies out of uncurated, real-world play data. We discuss failures of C-BeT on the knob tasks in Section 5.3. While our GoFAR baseline was able to move towards the task targets, it was unable to successfully grasp or interact with any of the target objects. We believe it may be the case because unlike the robot experiment in Ma et al. (2022), we do not have the underlying environment state, the tasks are much more complicated, and our dataset is an order of magnitude smaller ($400\,\mathrm{K}$ vs $45\,\mathrm{K}$).

**Behavior generation for longer horizons:** Next, we ask how well our models work for longer-horizon conditioning with multiple tasks. We choose play sequences from the dataset with multiple tasks completed and use their associated states as the conditions for our models. In our roll-outs, we calculate how many tasks completed in the original sequence were also completed in the conditional roll-outs. We calculate this metric over 3 conditioning sequences, and report the results in Table 5. We see that even without any high level controller, C-BeT is able to stitch together multiple tasks from play demonstrations to complete long-horizon goals.

**Generalization to prompt and environment perturbations:** A major requirement from any robot system deployed in the real world is to generalize to novel scenarios. We evaluate the generalizability of our learned policies in two different ways. In the first set of experiments, we collect fresh demonstrations that were not in the training set, and we condition our policies on such trajectories. We find that across the different tasks, even with unseen conditionings, C-BeT retains $67\%$ of the single-task performance, with $16/50$ task successes in total. In the second set of experiments, we add environmental distractors in the setup (Figure 1, bottom three rows) and run the single- and multi-task conditions on the modified environments. We see once again that the performance drops to around $67\%$ of original with two distractors on the scene, but if we keep adding (four or more) distractors, the robot is unable to complete any tasks.

## 5.3 ANALYSIS OF FAILURE MODES

We see a few failure modes in our experiments that may provide additional insights into learning from real-world play data. We discuss the most salient ones in this section.

**Failure in knob operation in the real world:** We see that in all of our real world experiments, the accuracy in operating the knob is consistently lower than all other tasks. This is due to the failure of the learned representations. Upon inspection of the dataset images' nearest neighbors in the representation space, we see that the BYOL-trained representation cannot identify the knob state better than random chance: the returned nearest neighbor differs in knob status often. Since the representation cannot identify the knob status properly, conditioning on it naturally fails.

**Importance of a multi-modal policy architecture:** One of our motivations behind incorporating the BeT architecture in our work is its ability to learn multi-modal action distributions. In our experiments, we show that for some single-task conditions such as opening the oven door, having no multi-modality is sufficient (Table 4), but for more complicated tasks and learning from a more interconnected form of play data, it is always the case that a multi-modal architecture prevents our policies from collapsing to sub-optimal solutions (Table 5).

# 6 RELATED WORK

**Outcome-conditioned behavior learning:** Behavior learning conditioned on particular outcomes, such as reward or goals, is a long studied problem (Kaelbling, 1993; Schaul et al., 2015; Veeriah et al., 2018; Zhao et al., 2019). Compared to standard behavior learning, learning conditioned behavior can generally be more demanding since the same model can be expected to learn a multitude of behaviors depending on the outcome, which can make learning long-term behavior harder (Levy et al., 2017; Nachum et al., 2018). As a result, a common line of work in outcome-conditioned learning is to use some form of relabeling of demonstrations or experience buffer as a form of data augmentation (Kaelbling, 1993; Andrychowicz et al., 2017; Ghosh et al., 2019; Goyal et al., 2022) similar to what we do in the paper. As opposed to goal or state conditioned learning, which we focus on in this paper, recently reward conditioned learning using a transformer (Chen et al., 2021) was introduced. However, later work found that it may not work as expected in all environments (Paster et al., 2022; Brandfonbrener et al., 2022) and large transformer models may not be necessary (Emmons et al., 2021) for reward conditioned learning. In this work, we find that using transformers is crucial, particularly when dealing with high dimensional visual observation and multi-modal actions.

**Learning from play data:** Our work is most closely related to previous works such as Lynch et al. (2019); Gupta et al. (2019), which also focus on learning from play demonstrations that may not be strictly optimal and uniformly curated for a single task. Learning policies capable of multiple tasks from play data allows knowledge sharing, which is why it may be more efficient compared to learning from demonstrations directly (Zhang et al., 2018; Rahmatizadeh et al., 2018; Duan et al., 2017; Pari et al., 2021; Young et al., 2021). Gupta et al. (2022) attempts reset-free learning with play data, but requires human annotation and instrumentation in the environment for goal labels.

**Generative modeling of behavior:** Our method of learning a generative model for behavior learning follows a long line of work, including Inverse Reinforcement Learning or IRL (Russell, 1998; Ng et al., 2000; Ho & Ermon, 2016), where given expert demonstrations, a model tries to construct the reward function, which is then used to generate desirable behavior. Another class of algorithms learn a generative action decoder (Pertsch et al., 2020a; Singh et al., 2020) from interaction data to make downstream reinforcement learning faster and easier, nominally making multi-modal action distribution easier. Finally, a class of algorithms, most notably Liu et al. (2020); Florence et al. (2021); Kostrikov et al. (2021); Nachum & Yang (2021) do not directly learn a generative model, but instead learn energy based models that need to be sampled to generate behavior, although they do not primarily focus on goal-conditioning.

**Transformers for behavior learning:** Our work follows earlier notable works in using transformers to learn a behavior model from an offline dataset, such as Chen et al. (2021); Janner et al. (2021); Shafiullah et al. (2022). Our work is most closely related to Shafiullah et al. (2022) as we build on their transformer architecture, while our unimodal baseline is a variant of Chen et al. (2021) that learns outcome conditioned instead of reward conditioned policy. Beyond these, Dasari & Gupta (2020); Mandi et al. (2021) summarizes historical visual context using transformers, and Clever et al. (2021) relies on the long-term extrapolation abilities of transformers as sequence models. The goal of C-BeT is orthogonal to these use cases, but can be combined with them for future applications.

# 7 DISCUSSION AND LIMITATIONS

In this work, we have presented C-BeT, a new approach for conditional behavior generation that can learn from offline play data. Across a variety of benchmarks, both simulated and real, we find that C-BeT significantly improves upon prior state-of-the-art work. However, we have noticed two limitations in C-BeT, particularly for real-robot behavior learning. First, if the features provided to C-BeT do not appropriately capture relevant objects in the scene, the robot execution often fails to interact with that object in its environment. Second, some tasks, like opening the oven door, have simpler underlying data that is not multimodal, which renders only meager gains with C-BeT. A more detailed analysis of these limitations are presented in Section 5.3. We believe that future work in visual representation learning can address poor environment features, while the collection of even larger play datasets will provide more realistic offline data for large-scale behavior learning models.

ACKNOWLEDGEMENT

We thank Sridhar Arunachalam, David Brandfonbrener, Irmak Guzey, Yixin Lin, Jyo Pari, Abitha Thankaraj, and Austin Wang for their valuable feedback and discussions. This work was supported by awards from Honda, Meta, Hyundai, Amazon, and ONR awards N00014-21-1-2758 and N00014-22-1-2773.

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

APPENDIX

## A SIMULATED ENVIRONMENT VISUALIZATIONS

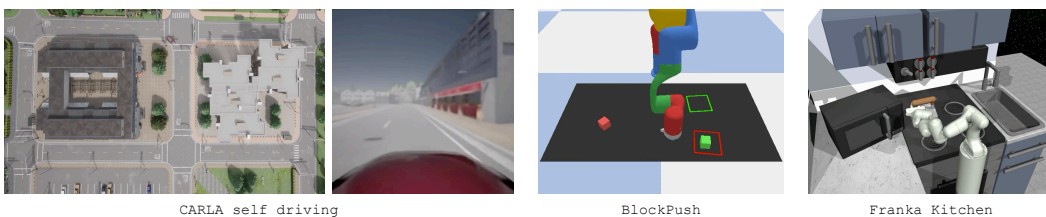

CARLA self driving                    BlockPush              Franka Kitchen

Figure 4: Visualizations of simulated environments that we evaluate our methods on, from left to right: CARLA self-driving (top down view and agent POV), BlockPush, and Franka Kitchen.

## B BEHAVIOR TRANSFORMERS

We use Behavior Transformers from Shafiullah et al. (2022) as our backbone architecture, building our conditional algorithm on top of it. In this section, we describe the BeT architecture and the training objective to help the readers understand the details of our algorithm.

### B.1 BeT ARCHITECTURE

BeT uses a repurposed MinGPT architecture to model multi-modal behavior. It uses the MinGPT trunk as a sequence-to-sequence model that tries to predict a sequence of actions $a_{t:t+h}$ given a sequence of states or observations $o_{t:t+h}$. Beyond just prediction the actions, however, BeT tries to model the multi-modal action distribtuion given the observations. To create a multi-modal model over the continuous action distribution, BeT uses an action encoder-decoder architecture that can encode each action vector into a discrete latent and a smaller-norm continuous offset. BeT does so by using an offline action dataset to create a $k$-means model of the actions. Then, an action is encoded into its associated bin out of the $k$-bins (binning), and a small continuous offset from the associated bin.

The BeT model, given a sequence of observations $o_{t:t+h}$, predicts a $k$-dimensional multinomial distribution over the $k$-bins, as well as a $k \times |A|$ dimensional matrix for offsets associated with each action bins. Sampling from the BeT action distribution is done via sampling a discrete bin first, taking its associated action offset, and then adding the bin center with the action offset.

### B.2 BeT TRAINING OBJECTIVE

Given an observation $o$ and its associated ground truth action $a$, we will now present the simplified version of how the BeT loss is calculated.

Let us assume the BeT model prediction is $\pi(o)_d \in \mathbb{R}^k, \pi(o)_c \in \mathbb{R}^{k \times |A|}$ for the discrete and the continuous parts of the predictions. Let us also assume that $\lfloor a \rfloor$ is the discrete bin out of the $k$ bins that $a$ belongs to, and $\langle a \rangle = a - \text{BinCenter}(\lfloor a \rfloor)$. Then, the BeT loss becomes

$$\mathcal{L}_{\text{BeT}} = L_{focal}(\pi(o)_d, \lfloor a \rfloor) + \lambda \cdot L_{MT}(\langle a \rangle, \pi(o)_c)$$

Where $L_{focal}$ is the Focal loss (Lin et al., 2017), a special case of the negative log likelihood loss defined as

$$\mathcal{L}_{focal}(p_t) = -(1 - p_t)^\gamma \log(p_t)$$

and $L_{MT}$ is the multi-task loss (Girshick, 2015) defined as

$$\text{MT-Loss}\left(\mathbf{a}, \left(\langle \hat{a}_i^{(j)} \rangle\right)_{j=1}^k\right) = \sum_{j=1}^k \mathbb{I}[\lfloor \mathbf{a} \rfloor = j] \cdot \|\langle \mathbf{a} \rangle - \langle \hat{a}^{(j)} \rangle\|_2^2$$

## C  IMPLEMENTATION DETAILS

### C.1  IMPLEMENTATION USED

In our work, we base our C-BeT implementation off of the official repo published at `https://github.com/notmahi/bet`. For the GCBC, WGCSL, and GoFAR baselines, we use the official repo released by the GoFAR authors `https://github.com/JasonMa2016/GoFAR/`.

### C.2  HYPERPARAMETERS LIST:

We present the C-BeT hyperparameters in Table 6 below, which were mostly using the default hyperparameters in the original Shafiullah et al. (2022) paper:

Table 6: Environment-dependent hyperparameters in BeT.

| Hyperparameter | CARLA | Block-push | Kitchen |
|---|---|---|---|
| Layers | 3 | 4 | 6 |
| Attention heads | 4 | 4 | 6 |
| Embedding width | 256 | 72 | 120 |
| Dropout probability | 0.6 | 0.1 | 0.1 |
| Context size | 10 | 5 | 10 |
| Training epochs | 40 | 350 | 50 |
| Batch size | 128 | 64 | 64 |
| Number of bins $k$ | 32 | 24 | 64 |
| Future conditional frames | 10 | 3 | 10 |

The shared hyperparameters are in Table 7.

Table 7: Shared hyperparameters for BeT training

| Name | Value |
|---|---|
| Optimizer | Adam |
| Learning rate | 1e-4 |
| Weight decay | 0.1 |
| Betas | (0.9, 0.95) |
| Gradient clip norm | 1.0 |

# D ROBOT ENVIRONMENT DEMONSTRATION TRAJECTORIES

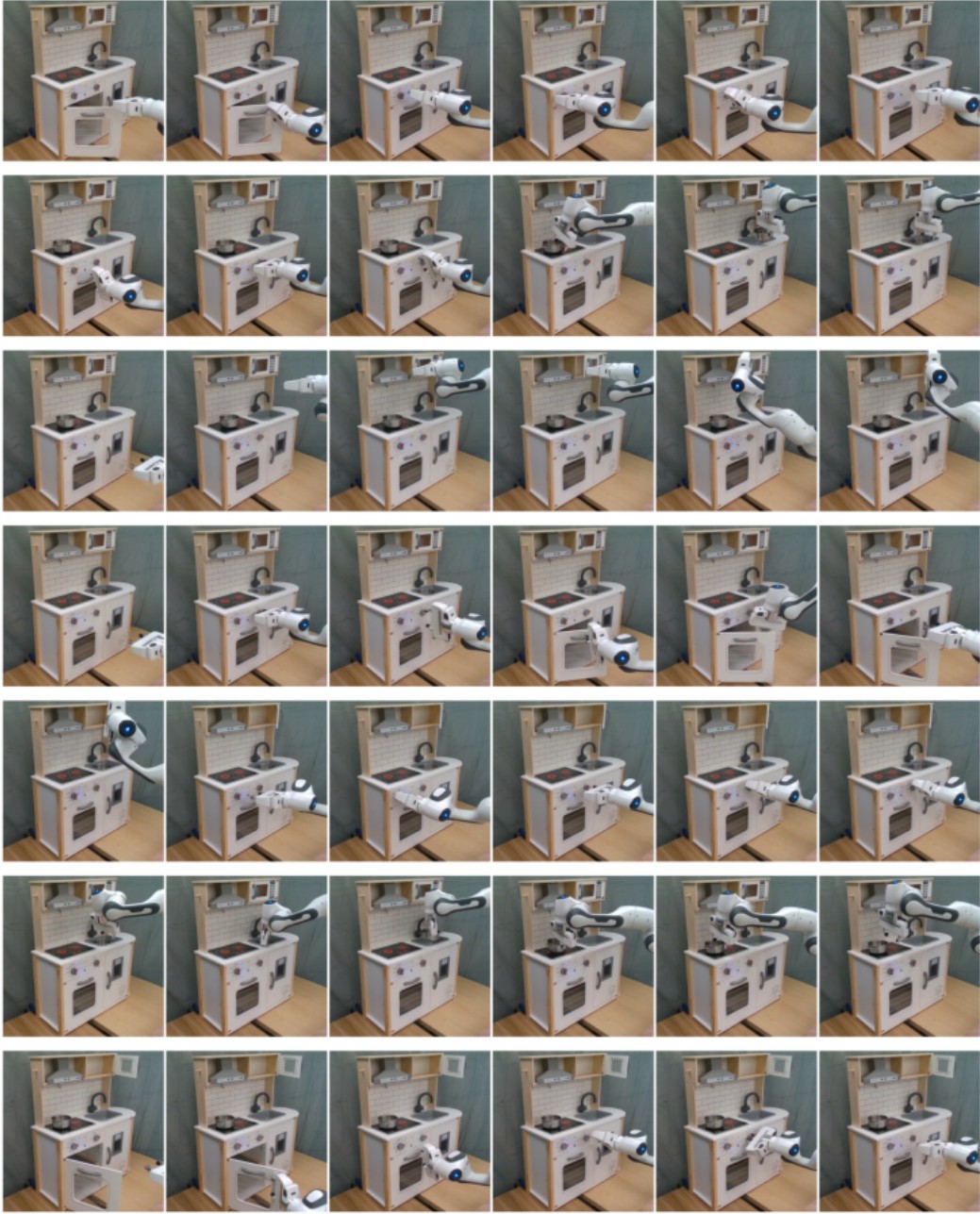

Figure 5: Sample demonstration trajectories for the real kitchen environment.

# E    SIMULATED ENVIRONMENT ROLLOUT TRAJECTORIES

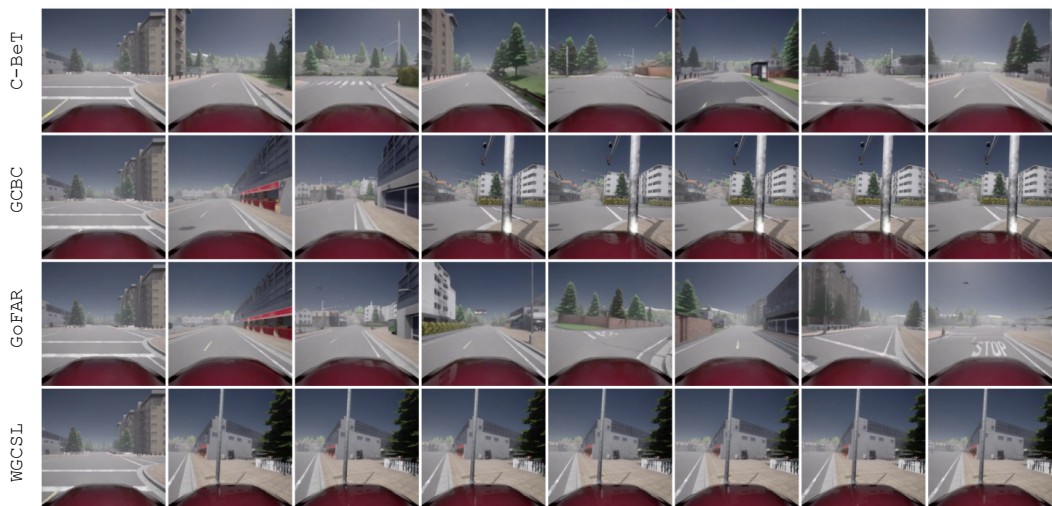

Figure 6: Sample demonstration trajectories for the CARLA self driving environment, conditioning on going to the right path.

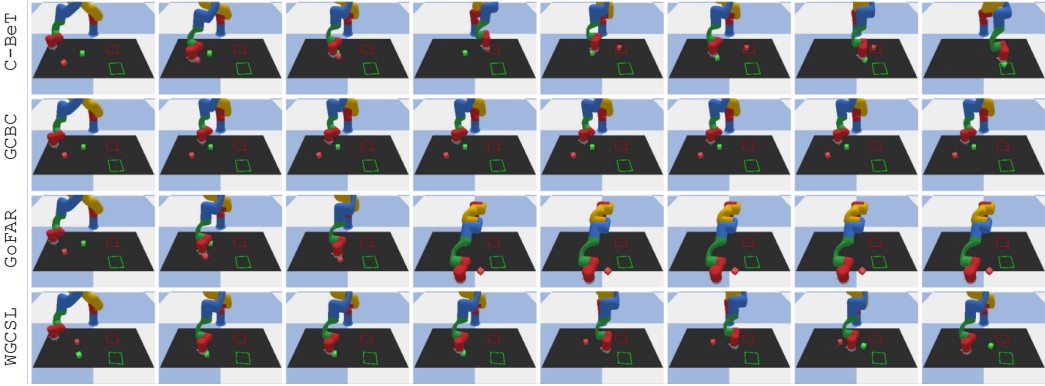

Figure 7: Sample demonstration trajectories for the multi-modal block pushing environment, conditioning on pushing the green block to green square and red block to red square.

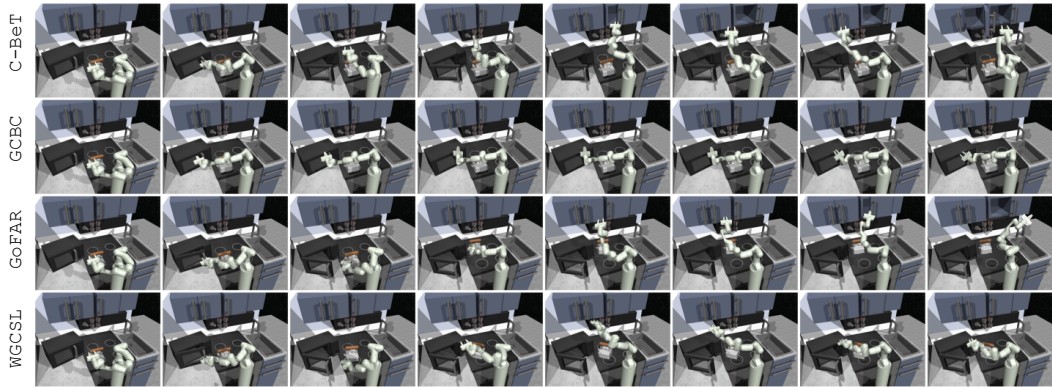

Figure 8: Sample demonstration trajectories for the Franka Kitchen environment, conditioning on completing the microwave, bottom knob, slide cabinet, hinge cabinet tasks.

