# OpenReview forum: "From Play to Policy: Conditional Behavior Generation from Uncurated Robot Data"
_ICLR.cc/2023/Conference — ICLR 2023 notable top 5%_

### Official Review · Reviewer_nm8H · 2022-10-22

**Confidence:** 4
**Correctness:** 3
**Technical Novelty And Significance:** 2
**Empirical Novelty And Significance:** 3
**Recommendation:** 6

**Clarity, Quality, Novelty And Reproducibility:**

The paper is well written with clarity and high quality. The code (conditional portion of BeT) is not included in the submission but the hyperparameters are specified in the appendix.



**Strength And Weaknesses:**

Strength:
- Goal-conditioned learning is a clean and simple formulation, compared to Q learning.
- C-BeT addresses the shortcomings of BeT by conditioning on goals to resolve ambiguities.
- This paper presents results in both simulated and real-world datasets and shows that C-BeT outperforms baselines in simulated benchmarks.

Weakness:
- The core idea is doing GCBC with BeT as the BC backbone, which is only incrementally novel.
- The criteria of choosing baselines is not super clear. A concurrent work of Play-LMP, RIL (Gupta et al.), that outperforms GCBC in the Kitchen domain isn’t included while GCBC is in. A concurrent work of GoFar, PLATO (Belkhale et al.), that outperforms Play-LMP isn’t included while WGCSL (which is shown to be not as good as GoFar) is included.
- The result metric for simulated tasks is confusing, what is the unit? Is it the number of successes over how many total tasks? Why not just use the standard success rate? Did you run multiple different seeds?
- The reason why BeT would outperform goal-conditioned methods isn’t well explained. It is a little counterintuitive and leads to the suspicion that the baseline models are not sufficiently tuned to work with the selected domains. Why not reuse the Play-LMP dataset/tasks? How would C-BeT and BeT perform in tasks without artificially introduced multimodal data?
- The type of generalization is very limited.
- The setup is fixed and the policy theoretically could have learned the door and knob tasks without the visual observation and be more robust to visual perturbations.
- The generalization test of “fresh” demonstrations are still of the same tasks.



References:

Gupta, Abhishek, et al. "Relay Policy Learning: Solving Long-Horizon Tasks via Imitation and Reinforcement Learning." Conference on Robot Learning. PMLR, 2020.

Belkhale, Suneel, and Dorsa Sadigh. "PLATO: Predicting Latent Affordances Through Object-Centric Play." arXiv preprint arXiv:2203.05630 (2022).




**Summary Of The Paper:**

This paper tackles the problem of offline learning from play data and presents a goal-conditioned version of behavior transformer (BeT) and demonstrates it outperforms several existing algorithms in selected simulation environments and a real robot setup.

**Summary Of The Review:**

This paper presents a conditional version of BeT and demonstrates that it performs well on a set of simulated tasks as well as a real robot kitchen environment. The novelty of the method is limited and the selection of baselines is not satisfying but I do appreciate the simplicity of this algorithm. I don’t recommend accepting this work in its current form but won’t argue strongly against it.

---

> ### Author Response · Authors · 2022-11-12
> **Response to Reviewer nm8H (2/2)**
>
> * **Limits of generalization:** You have mentioned that
> 	> The type of generalization is very limited… The setup is fixed and the policy theoretically could have learned the door and knob tasks without the visual observation and be more robust to visual perturbations… The generalization test of “fresh” demonstrations are still of the same tasks.
>
> 	To the best of our knowledge, C-BeT is the first work to learn conditional policies from uncurated and reward-free multi-modal data on a real robot. We see C-BeT as just the first step towards such a learning paradigm and agree that we have ways to go before reaching the generalization capabilities seen in other AI fields. From a practical perspective, as an academic lab we have to work within the constraints of our budget, which means learning from a small, 4.5 hour dataset and the lack of cross-scene, cross-task generalization that follows from that. But we are hopeful that with broader adoption, methods that build on C-BeT can scale to face harder generalization challenges.
> * **Use of vision in our policy:** We condition our model on visual goals, and specify long-horizon goals that cannot be expressed using a short proprioception sequence (for example, completing two tasks one after another). As a result, we know that our policy necessarily relies on visual input. We hope to add more experiments comparing purely proprioceptive or open loop baselines in the final version.
>
> Assuming our clarifications above have addressed your concerns, we would like to kindly ask whether you would be willing to update your score. If you have any further questions or clarifications to ask, we would be happy to respond as well.
>
> [1]  Gupta, Abhishek, et al. "Relay Policy Learning: Solving Long-Horizon Tasks via Imitation and Reinforcement Learning." Conference on Robot Learning. PMLR, 2020.
> [2]  Mandlekar, Ajay, et al. "Learning to generalize across long-horizon tasks from human demonstrations." arXiv preprint arXiv:2003.06085 (2020).
> [3]  Florence, Pete, et al. "Implicit behavioral cloning." Conference on Robot Learning. PMLR, 2022.
> [4]  Shafiullah, Nur Muhammad Mahi, et al. "Behavior Transformers: Cloning $ k $ modes with one stone." arXiv preprint arXiv:2206.11251 (2022).

---

> > ### Comment · Reviewer_nm8H · 2022-11-25
> > **a leap of faith**
> >
> > My main concern for this work is still that the technical novelty is incremental from the BeT paper while the empirical experiments are not extensive enough (which has improved over rebuttal by having more baselines, but the datasets used are still limited given empirical results are the main contribution).
> >
> > I do appreciate the idea of applying BeT to learning from play and the fact that it works almost out-of-the-box. I believe the robot learning community would be interested in using this work so won't argue against accepting this work. I am hoping the final version of this work would include more evaluations (or extensive discussion of limitations) and better treatment of the baselines.

---

> ### Author Response · Authors · 2022-11-12
> **Response to Reviewer nm8H (1/2)**
>
> We thank you for your comprehensive review of our work and for suggesting important baselines. We would like to request you to visit the [global comment](https://openreview.net/forum?id=c7rM7F7jQjN&noteId=BBZ3gfnRgn6) to see the results from the updated baselines, as well as our view about technical vs. empirical novelty. Our response to the rest of your questions and concerns are as follows:
>
> * **Additional baselines:** Thank you for suggesting RIL and PLATO as additional baselines. We have added comparison to RIL  (Gupta 2019) to our paper, as well as GTI (Mandlekar 2019), and a conditioned variant of IBC (Florence 2021) as suggested by other reviewers. However, we found that PLATO has stronger assumptions on the nature of play data collected (namely, that each demo trajectory has only one “pre-interaction”, “interaction”, and “post-interaction” phase, and that we are able to split each trajectory into those three phases) that is incompatible with our framework.
>
> * **Choosing baselines:** To clarify, we chose GoFAR as a baseline since
> 	* It represents a large area of current research, namely offline RL.
> 	* It is a peer reviewed publication appearing in NeurIPS 2022, and finally,
> 	* It has code published on github, so results are easily verifiable.
>    As for RIL, we initially did not consider it for two particular reasons:
> 	* The “high level policy” needs to predict “goal states” for the low level policy to follow, which may work for low-dimensional sim environments, but is hard to scale to high-dimensional image observations.
> 	* Moreover, according to (Gupta 2019) and (Shafiullah 2022), RIL is outperformed by the k-NN based LWR algorithm (2.56 > 2.4).
> * **Result metrics in simulation:** We are sorry for the confusion caused by our Table 2 results, where we followed the reporting convention in (Shafiullah et al. 2022). We have updated the caption to clarify them. But just as a recap, in CARLA, successfully reaching the goal counts for 1 reward, in Block Push, pushing a block in the target square counts for 0.5 reward (1 total, max), and in Kitchen completing each target task counts for 1 reward (4 total, max).
> We ran each task and each algorithm on 5 initializations and evaluated over different goals to measure performance over 100 rollouts in total, reporting the average over the 100 rollouts.
> * **Reasons why C-BeT outperforms GCBC:** We believe C-BeT’s performance improvement over GCBC is best understood through the lens of two major architectural changes:
> 	* Replacing MLP with minGPT, which can be seen by comparing the column GCBC with column C-BeT (unimodal).
> 	* Upgrading unimodal, MSE based action prediction to multi-modal modeling similar to (Shafiullah et al. 2022), which can be seen by comparing the column C-BeT (unimodal) with C-BeT (multimodal).
> * **Baseline performance:** We have tried to use publicly available code for the baselines whenever possible, scaling the architectures to have similar model capacity while regularizing to ensure that they do not overfit to the data. We have also published our own code to the supplementary material and in the anonymized website so that practitioners can benefit from it and readers can appreciate the minimal amount of tuning required to make C-BeT work in the real world. Hopefully the code release will allay your concerns about unfair comparison for C-BeT.
> 	> Why not reuse the Play-LMP dataset/tasks?
>
>  	Unfortunately they were never released publicly, which is why we use BeT datasets and tasks in our work.
> 	> How would C-BeT and BeT perform in tasks without artificially introduced multimodal data?
>
> 	We fail to understand what you might mean by *“artificial”* multi-modality, since the entire purpose of C-BeT is to learn from play-data in the real world (similar to that collected by our volunteer in the real world Franka kitchen), rather than unimodal expert demonstrations. In this context, the type of multi-modality we see in our datasets is rather natural.

---

> > ### Comment · Reviewer_nm8H · 2022-11-25
> > **comparison with baselines**
> >
> > I truly appreciate the authors for their detailed responses to my concerns. I am happy to see new baselines are added to the evaluation. Overall, I see the major contribution of this work as "why BeT outperforms existing goal-conditional backbones" and therefore would love to see a diverse set of baselines and a variety of dataset/tasks.
> >
> > For sim evaluations, it would be great if the authors could reproduce Table in the BeT paper (Shafiullah et al.). If the same evaluations are used, why are the numbers different? Are you using completely different datasets? It seems at least the dataset for the Kitchen tasks should be the same but the numbers in these tables do not match and it would be great if you can explicitly mention how you converted the numbers in BeT results into 1.77. Also, why is the max number of tasks 4 instead of 5 (did you remove one task)?
> >
> > When I say "artificial" multimodality, I am referring to the CARLA and block-push tasks. I appreciate the simplicity of these two tasks but they are artificially designed to have this multimodal aspect. It would be useful to see running C-BeT on an existing dataset, e.g., CARLA offers a demonstration dataset in the work of Codevilla et al. [1], that is more natural.
> >
> >
> > [1] Codevilla, Felipe, et al. "End-to-end driving via conditional imitation learning." 2018 IEEE international conference on robotics and automation (ICRA). IEEE, 2018.

---

> > > ### Author Response · Authors · 2022-11-26
> > > **Thank you, and clarification on metric**
> > >
> > > We would like to thank you for acknowledging our paper updates and for increasing your score.  We believe there has been a misunderstanding about table 2, which we would like to clarify along with addressing your other concerns.
> > >
> > > > For sim evaluations, it would be great if the authors could reproduce Table in the BeT paper (Shafiullah et al.). If the same evaluations are used, why are the numbers different?
> > >
> > > The table in (Shafiullah 2022) is the result of unconditional evaluation, whereas our evaluation is conditional. For the conditional evaluation, on CARLA or Block push, we fix targets, and on Kitchen we fix a set of tasks. Then we conditionally roll out the model being evaluated and count the overlap between the targets/goal tasks and the model’s achieved target or tasks. We repeat this process over different targets and goal tasks and report the average number as the conditional success rate. While in (Shafiullah 2022) BeT completing any task is a success, in our setup it has to complete one of the conditioned tasks to be considered a success. Since the original BeT algorithm does not have a way to take the goals in consideration, the numbers for BeT is lower across the board compared to (Shafiullah 2022), and is intended to be the “random” baseline that ignores the goal condition in our paper.
> > >
> > > > Also, why is the max number of tasks 4 instead of 5 (did you remove one task)?
> > >
> > > For Kitchen, in our evaluation, the max reward is 4 since we condition on states from the human demonstration which has a maximum of 4 tasks in any trajectory.
> > >
> > > >  When I say "artificial" multimodality, I am referring to the CARLA and block-push tasks… It would be useful to see running C-BeT on an existing dataset, e.g., CARLA offers a demonstration dataset in the work of Codevilla et al. [1], that is more natural.
> > >
> > > We tried to keep our evaluation tasks as close to (Shafiullah 2022) as possible, where CARLA is used more as an illustrative case for multi-modality. The Block push task is inspired by (Florence 2021), and while it's seemingly simple, both (Shafiullah 2022) and our work finds that traditional methods perform poorly in this task. However, we agree that having more diverse tasks would benefit the field of conditional behavior learning and we will appropriately discuss this in the final version of the paper.
> > >
> > >
> > > 1. Shafiullah, Nur Muhammad Mahi, et al. "Behavior Transformers: Cloning $ k $ modes with one stone." arXiv preprint arXiv:2206.11251 (2022).
> > > 2. Florence, Pete, et al. "Implicit behavioral cloning." Conference on Robot Learning. PMLR, 2022.

---

### Official Review · Reviewer_dNx6 · 2022-10-23

**Confidence:** 5
**Correctness:** 3
**Technical Novelty And Significance:** 2
**Empirical Novelty And Significance:** 2
**Recommendation:** 5

**Clarity, Quality, Novelty And Reproducibility:**

Clarity: good writing.
Quality: it's a solid execution of a conceptually simple method, although the choice of baselines for empirical evaluation is questionable.
Originality: questionable, as the method is a somewhat straightforward combination of two prior works.

**Strength And Weaknesses:**

Strengths:
+ I like the spirit of developing a unified algorithm for imitation learning from a variety of domains.
+ I like the real-robot experiment setup and appreciate the effort put into collecting the data and evaluating the policy on a physical robot platform.

Weaknesses:

- My first main concern about the paper is its technical novelty. As stated by the authors themselves, the paper is a somewhat straightforward combination of two prior works, Shafiullah et al., 2021 for the transformer architecture and the multi-modal action learning and Lynch et al., 2019 for goal conditioning. As much as I appreciate the comprehensive empirical study, I had a hard time justifying the technical novelty of this paper given that the main focus of the ICLR conference is on learning methods themselves.
- My second main concern is the baselines used in the paper. The strongest baselines are BET and Play-GCBC, but they are “set to fail” since the proposed method is the combination of the two methods. Given the baseline choices, I’m not exactly sure the point that the authors hope to convey through the experiment. Is the main contribution of the paper the transformer architecture and the multi-modal behavior policy proposed by BET? Because 5 out of the 6 baselines do not have these components. Is the main contribution of the paper goal-conditioning? Because the strongest baseline BET does not have goal conditioning and obviously will not be able to achieve the specified goal.
- There are also other missing baselines. For example, IRIS [1] and Mandlekar and Xu 2020 [2] are both goal-conditional imitation methods that focus on disentangling multimodal behaviors. Neither is mentioned nor compared. ImplicitBC (Florence et al., 2021) that aims to learn multi-modal behaviors can also be trivially adapted to be conditioned on goals, similar to how the authors adapted BET to be conditioned on goals.

[1] IRIS: Implicit Reinforcement without Interaction at Scale for Learning Control from Offline Robot Manipulation Data
[2] Learning to Generalize Across Long-Horizon Tasks from Human Demonstrations.



**Summary Of The Paper:**

The paper presents a method to learn from “play data” in a robot manipulation domain. The “play data” is first popularized by Lynch et al., 2018 and is defined as robot experience data collected through human teleoperation where the human teleoperator controls the robot to achieve a variety of goals in a domain. The difference between such data and, e.g., experience data from reinforcement learning is that neither the underlying goal nor reward signals is given. The proposed method aims to train a policy that (1) captures the multimodal behavior exhibited in such data (2) generate behaviors shown in the training data by conditioning on a desired goal state and (3) generalize to new behaviors not shown in the data. The method combines two prior works: Shafiullah et al. learns multimodal behaviors and is based on a transformer architecture. Lynch et al. (2019) makes the policy goal-conditional. The method is evaluated on a number of simulated domains, including CARLA and a table-top block pushing domain, and a real-world robotics setup where the robot is trained to manipulate objects in a toy kitchen. The paper shows that the proposed method outperforms a few baseline methods, including a prior works without using transformer architecture (Lynch 2019) and a transformer-based model that is not conditioned on the goal (Shafiullah et al., 2022).

**Summary Of The Review:**

As mentioned in the main comments, although I like the real robot evaluation setup, I have strong doubts about the novelty of the proposed method and the baseline choices in evaluation.

========================
Post-rebuttal review: I thank the reviewer for adding additional baseline results. At the same time, I still believe the research described in this manuscript has limited technical novelty. As much as I'm impressed by the empirical result on real robot experiment, I must consider this factor due to the nature of the venue. I have updated the review score to reflect this.

---

> ### Author Response · Authors · 2022-11-12
> **Response to Reviewer dNx6**
>
> We thank you for your detailed review, as well as the pointer to the important related work that we missed in our literature review. Please see our [global comment](https://openreview.net/forum?id=c7rM7F7jQjN&noteId=BBZ3gfnRgn6) for our view about technical vs. empirical novelty. In the following, we will try to address the rest of your concerns.
>
> * **Missing baselines:** We appreciate your pointers towards the relevant works. We have added adaptations of GTI and conditional IBC to our simulated baselines. We see that while they both learn a reasonable baseline, C-BeT outperforms them across all simulation tasks. Once the baselines are trained on the real robot, we would be happy to add their performance to the final paper as well. IRIS requires reward labels and thus is not compatible with our framework. Our updated list of baselines can be found in the updated manuscript, with the [results table in the global comment](https://openreview.net/forum?id=c7rM7F7jQjN&noteId=BBZ3gfnRgn6).
> * **Improvement over baselines:** First, we would like to clarify that wherever applicable, we have “modernized” the Play-LMP along with our newly added C-IBC and GTI baseline architectures by using transformer encoders as sequence models; thus the Play-LMP, C-IBC, GTI, and the C-BeT (unimodal)$^1$ baselines satisfy both of your goal-conditioned and transformer based criteria. Our contribution goes beyond merely adding a transformer into goal-conditioning, since we also show that modeling multi-modal actions is a necessity when it comes to modeling play demonstrations.
> We added BeT as a baseline to show the improvement over a strong but “random” policy to ablate and inform the reader about the improvement of properly adding goal-conditioning to it, which shows a consistent improvement on our simulated tasks and the real kitchen.
>
> Assuming our clarifications above have addressed your concerns, we would like to kindly ask whether you would be willing to update your score. If you have any further questions or clarifications to ask, we would be happy to respond as well.
>
> 1. Which can also be thought of as a goal-conditioned variant of decision transformer.
>
> [1] Corey Lynch, Mohi Khansari, Ted Xiao, Vikash Kumar, Jonathan Tompson, S. Levine, and Pierre Sermanet. Learning latent plans from play. Corl, 2019.
> [2] Leslie Pack Kaelbling. Learning to achieve goals. In IN PROC. OF IJCAI-93, pp. 1094–1098. Morgan Kaufmann, 1993.

---

### Official Review · Reviewer_DhxW · 2022-10-25

**Confidence:** 5
**Clarity, Quality, Novelty And Reproducibility:** Good.
**Correctness:** 4
**Technical Novelty And Significance:** 2
**Empirical Novelty And Significance:** 3
**Recommendation:** 8

**Strength And Weaknesses:**

*Strengths*

- Simple method building on BeT.
- Intuitive and well-motivated approach for the problem statement (BeT handles multimodality well, makes sense to use it for learning from play data)
- Strong results in simulation and on real hardware.

*Weaknesses*

- Limited novelty beyond BeT, basically just adding goal conditioning. But it's largely an empirical paper and the results are strong so not a big issue.

**Summary Of The Paper:**

This paper proposes a goal-conditioned transformer for learning from play data. The approach is a pretty straightforward combination of the Behavior Transformer (BeT) (Shafiullah et al) and goal-conditioning. BeT trains a transformer for imitation learning with a hybrid discrete-continuous action space and has been shown to be effective at capturing multimodal data. Learning from play data is known to be multi-modal, so it makes sense that a goal-conditioned BeT would be an effective way to learn from it.

The results are pretty strong, outperforming the relevant imitation and offline RL goal-conditioned baselines in a set of 3 simulation environments. Additionally, it has good results on a challenging real robot manipulation task.

**Summary Of The Review:**

Overall the paper extends the Behavior Transformer to be goal conditioned. Doing so enables effective goal-conditioned learning from play, outperforming relevant baselines on simulated and real robotics tasks.

---

> ### Author Response · Authors · 2022-11-12
> **Response to Reviewer DhxW**
>
> We thank you for your positive review. Please let us know if you have any additional questions and we will be happy to address them.

---

### Official Review · Reviewer_ySFW · 2022-11-01

**Confidence:** 5
**Correctness:** 3
**Technical Novelty And Significance:** 3
**Empirical Novelty And Significance:** 3
**Recommendation:** 8

**Clarity, Quality, Novelty And Reproducibility:**

I found this paper incredibly clear and easy to read; the quality and thoroughness of the experiments (and attention to baselines) is commendable, and I really buy these results. I like that the authors chose to evaluate on open-source simulated environments as well as a real-robot; it speaks a lot to the reproducibility of this work!

Minor: Seems like Claim 2 in the introduction “C-BeT represents the first work to demonstrate that competent visual policies for real-world tasks can be learned from reward-free play data” isn’t quite right; Implicit BC learns from play-esque multimodal data without rewards, as does follow-up work like Implicit Kinematic Policies (but I could be wrong; might just be a wording thing around the distinction between “play” and “multimodal demos”).


**Strength And Weaknesses:**

I think this is a very strong paper, showing that existing Behavior Transformers can scale to multimodal, reward free play data. In general, I really believe in the results in this paper, and am excited for the possibilities of using such an approach for general offline policy learning on real (and simulated) robotic tasks.

The weaknesses of this work are generally just weaknesses around the feasibility of visual goal-conditioning in the wild. In the real world, it’s not clear whether (given a new task, or a combinatorial explosion of objects/states) we’ll be able to provide robots with meaningful “goal images” that capture what we want to happen; for example, across all the tasks in this work, only a handful of object positions change from start state to goal state; in heavily cluttered environments with dynamic objects, it feels like such an approach would really suffer (in sample efficiency and maybe in just general applicability). This is somewhat shown in the videos, where we add random objects (though that’s also just because the environment is so out of distribution).


**Summary Of The Paper:**

This work presents a straightforward, but impressive extension of prior work on “Behavior Transformers” for learning policies from offline, reward-free data, by introducing the ability to learn goal-conditioned policies from undirected play data. Crucially, the play data used to learn policies in this work in inherently multimodal (and in some cases suboptimal); yet, using the goal conditioned behavior transformer (C-BeT) approach, we’re able to overcome such problems — *without any online exploration*.

The proposed approach builds on a history of prior work in goal-conditioned behavior cloning from visual observations, and makes a simple tweak to the underlying Behavior Transformer (think a causal transformer that eats states, and outputs actions) architecture, by additionally conditioning on target goal observations (images) of where we’d like the environment to end up. This is a powerful objective, and across three complex simulated environments (CARLA self-driving, multimodal block pushing, and the Franka Relay Kitchen environment) — each with varying levels of multimodality/suboptimality, C-BeT shows itself to be extremely strong.

However, the truly impressive part of this work is the real-robot evaluation; from just 4.5 hours of real user “play” on a Franka robot, C-BeT is able to learn 5 complex manipulation tasks in a “toy kitchen” to a meaningful level of competency — far better than other approaches from the offline RL literature like GO-FAR. In general, I found the evaluation to be thorough, ablations comprehensive, and comparisons to prior art meaningful!

**Summary Of The Review:**

I think this is a straightforward and impactful extension of prior work on behavior transformers. I think that being able to learn robust goal-conditioned policies from uncurated reward-free play data in a manner that is 1) simple and 2) completely offline is very meaningful, and a clear win for the field.

I really like this work!

---

> ### Author Response · Authors · 2022-11-12
> **Response to Reviewer ySFW**
>
> First of all, we thank you for your enthusiastic review – it’s heartening that you find our work “impressive” and “impactful”, and our paper very strong. Let us respond to the concerns you have mentioned in your review:
> * **Claims about learning from play:** We would like to draw a distinction between multi-modal demonstrations and play data in the sense that while we believe play data is almost always multi-modal, not all multi-modal demonstrations count as play data. However, we see how our claim can be misunderstood, so we have updated the statement in our paper to remove the line about being the first. We hope you find it more appropriate.
> * **Effectiveness of image-conditioning:** We agree that a lot of work still remains in finding the best representation on which we can train conditional models like C-BeT. However, the flexible architecture of C-BeT should be able to fit that representation, whether it encodes each frame of observation as a single vector or a sequence of object representations or semantic tokens.
>
> Please let us know if you have any more questions. We will be happy to address them.

---

> > ### Comment · Reviewer_ySFW · 2022-11-18
> > **Thanks**
> >
> > Thanks for addressing these; I also saw the results adding the new baselines (especially the conditional IBC baseline), and find this work to be very strong. I'll be advocating for this paper!

---

### Author Response · Authors · 2022-11-12
**General response to the reviewers, and list of updates**

Thank you for your thoughtful feedback and for finding our paper impressive (ySFW) and strong (DhxW). Concurrently, concerns about the novelty of this work and baselines have also been raised by the reviewers. In this global comment we would like to clarify the novelty of this work, followed by summarizing the changes we have made in the paper.

**Concerning technical vs. empirical novelty:** There are a myriad of papers that seek to learn policies on play-style data, however few prior work are able to learn conditional skills on uncurated real-world robotic data. The key novelty of this work is to show that these real-world skills can be learned from high-dimensional observations. We agree that in terms of technical novelty this work is a simple combination of existing ideas. However, we see this simplicity as a strength rather than a weakness. While reviewers ySFW and DhxW seem to agree with our sentiment and the value of our real-world robotic results, we hope that the other reviewers can also appreciate these results.

We have also updated our manuscript with the following additional experiments and modifications to incorporate your feedback into our work:

* **Additional baselines:** On the suggestion of reviewers, we have added three new baselines –  a conditional variant of IBC (Florence et al. 2021) mentioned by reviewer ySFW and dNx6, Relay Imitation Learning (Gupta et al. 2019) suggested by reviewer nm8H and a variant of Generalization Through Imitation (GTI)  (Mandlekar et al. 2019) as recommended by reviewer dNx6. While they present a strong set of baselines, even the best of them are outperformed by C-BeT by 1.5-3x in different environments. The updated Table 2 can be seen here:

> | Task 	| GCBC 	| WGCSL 	| GoFAR 	| Play-LMP 	| RIL 	| C-IBC 	| GTI 	| BeT 	| C-BeT (unimodal) 	| C-BeT (multimodal) 	|
|----|----|---|---|---|---|---|---|---|---|---|
| CARLA 	| 0.04 	| 0.02 	| 0.31 	| 0.0 	| 0.59 	| 0.65 	| 0.74 	| 0.72 	| 0.62 	| **0.98** 	|
| BlockPush 	| 0.06 	| 0.10 	| 0.04 	| 0.02 	| 0.03 	| 0.01 	| 0.03 	| 0.34 	| 0.31 	| **0.51** 	|
| Kitchen 	| 0.74 	| 1.17 	| 1.24 	| 0.04 	| 0.39 	| 0.13 	| 1.61 	| 1.77 	| 2.74 	| **2.80** 	|

* **Better explanation for our metric:** According to a suggestion made by reviewer dNx6, we have updated the caption for table 2 with a clearer explanation of the metric. Generally, we followed the convention in (Shafiullah et al. 2022), which should hopefully be clearer with the new caption.
* **Code release:** We have released our anonymized code in the supplementary materials, as well as in our anonymized website [https://cbet-anon.github.io](https://cbet-anon.github.io) for reproducibility.

We hope that these updates to our paper inspire further confidence in our work. At the same time, we invite any further questions or feedback that you may have on this paper.

[1]  Florence, Pete, et al. "Implicit behavioral cloning." Conference on Robot Learning. PMLR, 2022.
[2]  Gupta, Abhishek, et al. "Relay Policy Learning: Solving Long-Horizon Tasks via Imitation and Reinforcement Learning." Conference on Robot Learning. PMLR, 2020.
[3]  Mandlekar, Ajay, et al. "Learning to generalize across long-horizon tasks from human demonstrations." arXiv preprint arXiv:2003.06085 (2020).
[4]  Shafiullah, Nur Muhammad Mahi, et al. "Behavior Transformers: Cloning $ k $ modes with one stone." arXiv preprint arXiv:2206.11251 (2022).

---

### Decision · Program_Chairs · 2023-01-20

**Decision:**

Accept: notable-top-5%

**Justification For Why Not Higher Score:**

N/A

**Justification For Why Not Lower Score:**

I believe that there is significant value in demonstrating that goal-conditioned learning from play is a powerful approach that can be solved by combining a simple algorithm with the already existing behavior transformer architecture.  I honestly don't see this as incremental given that the learning from play setting is potentially considerably more broadly applicable than the more narrow supervised imitation learning setting.

**Metareview: Summary, Strengths And Weaknesses:**

This work introduces Conditional Behavior Transformers, an approach for conditioning a robot control policy on desired future observations in order to support learning from diverse "play" data.  The resulting conditional policies can be prompted with target observations or demonstration trajectories and, assuming they generalize well, will be able to execute behaviors to achieve the targets.  This approach is demonstrated on simulation and laboratory-based real-world settings.

This paper was met with strongly conflicting initial opinions.  Reviewers ySFW and DhxW found the work straightforward, but impressive due to strong empirical results that worked on a real robot platform.  On the other hand, Reviewers dNx6 and nm8H found the work simple and effective, but too straightforward (i.e., too obvious) and too incremental.


**Note From Pc:**

if the above contains the word "oral" or "spotlight" please see: "oral" presentation means -> notable-top-5% and "spotlight" means -> notable-top-25%. As stated in our emails, we are disassociating presentation type from AC recommendations

**Summary Of Ac-Reviewer Meeting:**

We had a video chat discussion.  Unfortunately, reviewer ySFW did not attend the meeting, so the AC and 3/4 reviewers participated.

Brief summary of major discussion points that arose:
- Reviewer nm8H reiterated the work was too incremental, too limited technical contribution.
- Reviewer DhxW agreed that technical novelty is limited, but the empirical evaluation is impressive and it essentially seems state-of-the-art compared to reasonable baselines.  The main thing that matters is that the idea works well.
- Reviewer dNx6 acknowledged a difference in review philosophy.  Focused on what can be learned from the paper.  Agreed this work is useful as a system integration effort (considered it possibly suitable for a robotics venue such as RSS or ICRA).   Satisfied with author response about baselines, but unsure how to isolate the technical novelty.
- Reviewers nm8H and dNx6 agreed with each other that the behavior transformer is the core thing that makes it work and while they could see themselves building on that in future work, this paper doesn't contain a big enough innovation.

In moderating the video chat discussion, after hearing the above opinions, I (the AC) did bring up that I leaned positive based on the impressiveness of the empirical results.  Personally, I find it hard to accept arguments about things being obvious in retrospect.  However, this depends on whether the achievement is sufficiently impressive as to be considered big.

Ultimately, the discussion settled into a semi-consensus that hinges on whether it is obvious that behavior transformers would be applicable to goal-conditioned learning from play given that they were previously shown to work on learning directly from demonstrations.  We discussed how distinct learning from play vs supervised imitation learning are as problem settings and we discussed whether goal-conditioning was too limited of a modification.

After the video discussion, in light of author updates, reviewer nm8H updated their score from a 5-->6 and reviewer dNx6 updated their score from a 3-->5.